# Single-Cell Network-Based Drug Repositioning for Discovery of Therapies against Anti-Tumour Necrosis Factor-Resistant Crohn’s Disease

**DOI:** 10.3390/ijms241814099

**Published:** 2023-09-14

**Authors:** Min Seob Kwak, Chang-Il Hwang, Jae Myung Cha, Jung Won Jeon, Jin Young Yoon, Su Bee Park

**Affiliations:** 1Department of Internal Medicine, Kyung Hee University Hospital at Gangdong, College of Medicine, Kyung Hee University, Seoul 05278, Republic of Korea; 2Department of Microbiology and Molecular Genetics, College of Biological Sciences, University of California Davis, Davis, CA 95616, USA; cihwang@ucdavis.edu

**Keywords:** Crohn’s disease, anti-TNF, drug repositioning, single cell sequencing, refractory disease

## Abstract

Primary and secondary non-response affects approximately 50% of patients with Crohn’s disease treated with anti-tumour necrosis factor (TNF) monoclonal antibodies. To date, very little single cell research exists regarding drug repurposing in Crohn’s disease. We aimed to elucidate the cellular phenomena underlying resistance to anti-TNF therapy in patients with Crohn’s disease and to identify potential drug candidates for these patients. Single-cell transcriptome analyses were performed using data (GSE134809) from the Gene Expression Omnibus and Library of Integrated Network-Based Cellular Signatures L1000 Project. Data aligned to the Genome Reference Consortium Human Build 38 reference genome using the Cell Ranger software were processed using the Seurat package. To capture significant functional terms, gene ontology functional enrichment analysis was performed on the marker genes. For biological analysis, 93,893 cells were retained (median 20,163 genes). Through marker genes, seven major cell lineages were identified: B-cells, T-cells, natural killer cells, monocytes, endothelial cells, epithelial cells, and tissue stem cells. In the anti-TNF-resistant samples, the top 10 differentially expressed genes were *HLA-DQB-1*, *IGHG1*, *RPS23*, *RPL7A*, *ARID5B*, *LTB*, *STAT1*, *NAMPT*, *COTL1*, *ISG20*, *IGHA1*, *IGKC*, and *JCHAIN*, which were robustly distributed in all cell lineages, mainly in B-cells. Through molecular function analyses, we found that the biological functions of both monocyte and T-cell groups mainly involved immune-mediated functions. According to multi-cluster drug repurposing prediction, vorinostat is the top drug candidate for patients with anti-TNF-refractory Crohn’s disease. Differences in cell populations and immune-related activity within tissues may influence the responsiveness of Crohn’s disease to anti-TNF agents. Vorinostat may serve as a promising novel therapy for anti-TNF-resistant Crohn’s disease.

## 1. Introduction

Crohn’s disease (CD) is a chronic inflammatory condition affecting the entire gastrointestinal tract; it has a variable disease course and wide spectrum of severity. Thus, it greatly influences the quality of life [1,2,3].

Although the exact pathogenesis of CD is not completely understood, an aberrant immune response driven by innate immunity triggers the release of proinflammatory mediators, such as tumour necrosis factor-alpha (TNF-α) and interleukin, resulting in not only the activation of the adaptive immune system but also tissue damage [4,5,6].

Currently, non-biological conventional drugs, including 5-aminosalicylic acid, corticosteroids, and the purine analogues azathioprine and 6-mercaptopurine, remain the mainstay of treatment for CD. However, for the last few decades, biologic agents, such as anti-TNF-α inhibitors, have exhibited successful treatment outcomes and have thereby changed the paradigm of CD treatment.

However, although TNF-α is a key factor in immune-mediated inflammatory diseases, therapeutic strategies maximizing the benefit to most CD patients are currently unavailable owing to its complex aetiology [7]. Therefore, many patients experience primary non-response to these agents or a loss of response over time, which consequently lead to disease flare-ups during scheduled maintenance therapies in CD [8].

Accordingly, other therapeutic options with different mechanisms of action that can induce clinical remission in patients with anti-TNF-refractory CD are required.

In the present study, we aimed to investigate developmental features resulting from the heterogeneity of cell populations at the cellular level constituting the diseased tissue using single-cell network biology and to identify suitable drug candidates for patients with CD resistant to anti-TNF therapy. 

## 2. Results

### 2.1. Extracting Single-Cell Transcriptomic Profiles

In total, 93,893 cells were retained for biological analysis, with a median of 20,163 genes. After normalization of gene expression and PCA, we used graph-based clustering to partition the cells into 14 clusters, indicating the underlying biological differences (Figure 1A). Through marker genes, seven major cell lineages were identified: T-cells (49,609 cells, 52.8%, marked with IL7R, CCR7, CD3D, CD3E, and CD3G); B-cells (28,711 cells, 30.6%, marked with CD79A, CD74, IGHM, MS4A1, and CD19); natural killer (NK) cells (4706 cells, 5.0%, marked with GNLY and NKG7); monocytes (4590 cells, 4.9%, marked with CD14); tissue stem cells (2395 cells, 2.6%, marked with ATP4A); epithelial cells (2070 cells, 2.2%, marked with EPCAM, KRT18, and KRT8); and endothelial cells (1812 cells, 1.9%, marked with PECAM1, CD34, CDH5, VWF, and CDH5) (Figure 1B). 

### 2.2. Immune Cellular Heterogeneity between Different Anti-TNF Responsiveness

The proportion of each cell cluster within the intestine after anti-TNF therapy varied significantly between the two groups (inflamed tissue vs. normal tissue) (Figure 2A,B).

Compared with normal tissue, monocytes and B-cells in inflamed tissue were significantly increased (log-fold changes of 0.173 and 0.092, respectively; *p* < 0.001; Appendix A). However, the proportions of T-cells (log FC of −0.031; *p* < 0.001), NK cells, and endothelial cells significantly decreased (log FC of −0.123; *p* < 0.001), as did those of epithelial cells (log FC of −0.223; *p* < 0.001) and tissue stem cells (log FC of −0.224; *p* < 0.001).

### 2.3. Transcriptional Landscape Heterogeneity of Cells Associated with Anti-TNF Therapy Resistance 

To understand the biological differences in anti-TNF responsiveness, we identified the genes that were highly differentially expressed between the two groups. A total of 7679 genes were identified as differentially expressed between tissues with and without anti-TNF responsiveness to CD. Specifically, we identified 2268, 1706, and 1697 DEGs in endothelial, tissue stem, and epithelial cells across the cell types, respectively. The following cells showed a relatively smaller number of genes that were differentially expressed between the two groups: 918 DEGs in monocytes, 558 DEGs in T-cells, 382 DEGs in NK cells, and 150 DEGs in B-cells. In anti-TNF-resistant samples, the top 10 DEGs were *HLA-DQB-1*, *IGHG1*, *RPS23*, *RPL7A*, *ARID5B*, *LTB*, *STAT1*, *NAMPT*, *COTL1*, and *ISG20*; *DUSP1*, *FABP6*, *SPRY1*, *RGS1*, *CITED2*, *CD96*, *KLRD1*, *PRMT9*, *PPP1R15A*, and *RGCC* were significantly highly expressed in normal tissues (Figure 3). A group of distinct DEGs was identified for each of the seven cell lineages (Figure 4). We found that *IGLC2*, *JCHAIN*, *IGKC*, and *IGHA1* were robustly distributed in all cell lineages, mainly in B-cells (Figure 4).

### 2.4. Functional Analysis and Drug Repositioning 

We then performed GO functional enrichment analysis on the overrepresented genes to compare the transcriptomes of normal and inflamed tissues (Figure 5).

Through molecular function analyses, we found that the biological functions of both the monocyte and T-cell groups mainly involved the immune-mediated function category, which was consistent with the results from both edgeR and limma analyses (Figure 5). Monocyte lineages were strongly enriched for chemokine activity, chemokine binding activity, cytokine activity, cytokine receptor binding, and immune receptor activity; in contrast, in T-cells, these signals were relatively diminished, but the associated genes functioned more widely for immune-related activity (Figure 5).

Therapeutic drug scores were estimated using the significance of reversing the differential gene expression pattern based on consistent DEGs and cell type proportions from the biological pathway. Using multi-cluster drug repurposing predictions, 626 drugs were predicted for patients with anti-TNF-refractory CD. Among them, vorinostat was identified as the top drug candidate (FDR < 0.05, overall drug score > 0.99 quantiles) for patients with anti-TNF-refractory CD (Table 1 and Appendix A).

## 3. Discussion

To the best of our knowledge, this is the first study to identify potential drug candidates by evaluating the biological processes underlying anti-TNF therapy resistance, using single-cell profiling in patients with CD. 

Researchers have suggested that anti-TNF resistance in patients with CD may arise through various biological mechanisms. A recent German study suggested that augmented IL-23 production by CD14+ monocyte/macrophages within the intestinal mucosa blocks anti-TNF-induced apoptosis during treatment, thereby inducing apoptosis-resistant intestinal TNFR2 + IL23R+ T-cells in CD [9]. Another experimental study revealed that activated STAT3 drives T-cell resistance against apoptosis in CD [10]. These studies suggest that the activation or expansion of immune cells, such as monocytes/macrophages and T-cells, in anti-TNF non-responders confers resistance against T-cell death via IL-23. Our previous results demonstrated that anti-TNF therapy resistance in patients with CD may be attributed to low immune cell activation due to the differential gene expression status of IL-17A/IL-17F produced by CD4+ T-cells [11]. 

In summary, T-cells and monocyte/macrophage lineages within the intestinal mucosa of patients with CD play a central role in tuning the response to anti-TNF therapy via cytokines and co-stimulatory signals.

From this perspective, we investigated the molecular variability at the cellular level for anti-TNF resistance in CD, and the results were in agreement with this concept.

Our results show higher B-cell and monocyte lineages and, thereby, identify the differences in immune-mediated pathways within tissues as key microenvironmental signatures for anti-TNF therapy resistance in CD. Interestingly, the actual transcript levels of T-cells were significantly higher in anti-TNF-resistant samples despite the relatively small number of T-cells. 

This may be the basis of mechanism-based target identification for drug repurposing. Additionally, Martin et al. showed that a prominent immune feature consisting of B-cells and inflammatory monocytes/macrophages correlated with anti-TNF resistance in tissue samples of CD, and hypothesised that a unique module consisting of stromal cells, monocytes/macrophages, activated T-cells, and B-cells, not just one type of cell lineage, may influence the responsiveness to anti-TNF agents [12]. 

On the basis of these results, we investigated potential drug candidates targeting key signalling pathways, by analysing genes expressed by the derived major cell lineages using a computational approach.

We estimated the drug score, which predicts drug efficacy, using the proportions of seven cell types, the significance of reversed differential gene expression patterns, and the ratio of reversed significantly deregulated genes over single-cell clusters associated with anti-TNF resistance. We found that vorinostat has a strong therapeutic potential to reverse the immune response within T-cell clusters, with the highest drug score across all patients.

Therefore, vorinostat is a potential novel therapeutic option for the management of patients with anti-TNF-resistant CD. 

Vorinostat, a histone deacetylase inhibitor (HDACi), is an anti-cancer agent that has recently been reported to modulate the immune system; however, the mechanisms underlying its activity are largely unknown [13,14]. 

Several studies have demonstrated that vorinostat ameliorates inflammation by reducing monocyte activation and modulating T-cell immune responses. Vorinostat has been shown to inhibit the differentiation, maturation, and endocytosis of human CD14(+) monocyte-derived dendritic cells and further inhibit their stimulation of allogeneic T-cell proliferation, thereby ameliorating experimental autoimmune encephalomyelitis [15]. Another study has suggested that vorinostat may considerably inhibit monocyte/macrophage activity by repressing Th1 and Th17 cells and reducing the TNF-α level in autoimmune diseases [16]. The results of graft versus host studies have demonstrated that vorinostat has a strong T-cell-suppressive effect [17].

A recent Australian study demonstrated that B-cell function is particularly sensitive to HDACi modulation. In vitro, HDACis are capable of altering diverse aspects of the B-cell response in an HDACi-class-dependent manner that is maintained for both T-cell-dependent and T-cell-independent activation [14]. 

In this study, we provide comprehensive insights into cell-type-specific gene regulation across different levels of anti-TNF responsiveness via single-cell transcriptome analysis, which allows for the detailed analysis of cell-to-cell variability in gene expression. Furthermore, our concordantly expressed gene-based drug repurposing approach presents a higher probability of identifying promising repurposed drugs, thereby enabling the advancement of personalized therapy for CD. 

Nevertheless, our study has certain limitations. First, the sample size is relatively small. To overcome this problem, a dropout experiment method within the algorithm was applied to calculate the drug score. Second, our in silico findings alone, without experimental validation, do not fully explain the biological mechanisms underlying the responsiveness to anti-TNF-α therapy in patients with CD. However, our findings are consistent with cumulative experimental evidence, and they may have relevant implications for the prediction of treatment response to anti-TNF agents in CD. Furthermore, our findings may facilitate a more accurate stratification of patients with CD through evaluation of the immune microenvironment before and/or during anti-TNF therapy.

In conclusion, our results reveal that the modulation of T-cell and monocyte lineages might serve as a promising approach to achieve an optimal response in anti-TNF-refractory patients with CD. Vorinostat, an HDACi inhibitor, may be a novel therapeutic option for patients with anti-TNF-refractory CD. However, further prospective, detailed experimental studies are required for verifying these findings.

## 4. Materials and Methods

### 4.1. Description of Datasets

The datasets analysed in this study were retrieved from the National Center for Biotechnology Information GEO database (http://www.ncbi.nlm.nih.gov/geo/, accessed on 24 August 2023). We analysed the single-cell RNA sequencing profiles of the ileal specimens of 22 patients with anti-TNF-refractory CD from the GSE134809 dataset, which included information from 11 inflamed tissues and 11 noninflamed adjacent tissues. 

### 4.2. Processing of Single-Cell RNA Sequencing Data

Data aligned to the Genome Reference Consortium Human Build 38 (GRCh38) reference genome using the Cell Ranger software (v4.0.2) [18] were processed using the Seurat package [19]. We first generated Seurat Objects using all gene expression matrices and filtered them to exclude cells with fewer than 200 genes, more than 6000 genes, and more than 10% mitochondrial genes, and then constructed a combined Seurat Object. After normalization and scaling of the merged Seurat Objects, the top 2000 highly variable features were selected for further clustering analysis. To reduce dimensionality, principal component analysis (PCA) using the Uniform Manifold Approximation and Projection (UMAP) algorithm was performed based on the features, and t-distributed stochastic neighbour embedding (t-SNE) was used for visualisation of the top 15 principal components.

### 4.3. Identification of Marker Genes and Enrichment Analysis

The transcriptional markers of cell clusters were identified using the Seurat ‘FindMarkers’ function. Using both limma and edgeR analyses, the differentially expressed genes (DEGs) between the putative clusters were selected as those with a percentage of cells with an expression higher than 0.25, an average log fold change (log FC) larger than 0.25, and adjusted *p* values less than 0.05. 

Cell types for each cluster were annotated using canonical marker genes, and cell cycle phase-specific changes were identified in different cell clusters based on the cell cycle score.

To allow for cell cycle phase assignment at all stages of the cell cycle, a mitotic cell population was identified. To capture the significant functional terms, gene ontology (GO) functional enrichment analysis was performed on the marker genes, considering a threshold-adjusted *p*-value of <0.01.

### 4.4. Drug Repurposing

To identify potential candidate drugs for repurposing, we used the ASGARD package [20], which represents the effect of reversed differential gene patterns of the drugs/compounds, by linking to the Library of Integrated Network-Based Cellular Signatures (LINCS) L1000 project [21]. The large intestine-specific drug score across all clusters of cells was calculated at the individual sample level; this enabled the prediction of the efficacy of the captured drug/compound using the cell type proportion, the significance of the reversed DEG pattern, and the ratio of reversed significantly deregulated genes over resistance-related single-cell clusters. 

### 4.5. Statistical Analyses

The *p*-values < 0.05 were considered to indicate statistical significance. All data processing and statistical analyses were performed using R software (version 4.0.5) and Python (version 3.7.1). The analyses were run on a server with an Intel Xeon processor (2 × Six-Core), 128 GB memory, and two-GPU Nvidia TITAN X. Ethics approval was obtained from the Institutional Review Board of the Kyung Hee University Hospital at Gangdong, Seoul, Republic of Korea (KHNMC IRB 2023-02-016). 

## Figures and Tables

**Figure 1 ijms-24-14099-f001:**
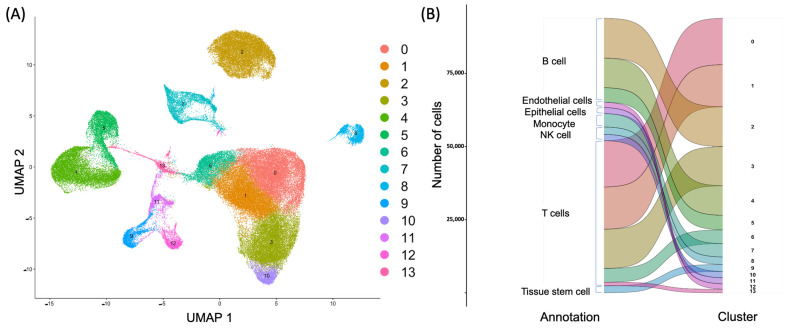
Unsupervised clustering of cells (**A**) and automatic annotation of cell types (**B**) using the UMAP dimensionality reduction technique in ileal tissues in CD.

**Figure 2 ijms-24-14099-f002:**
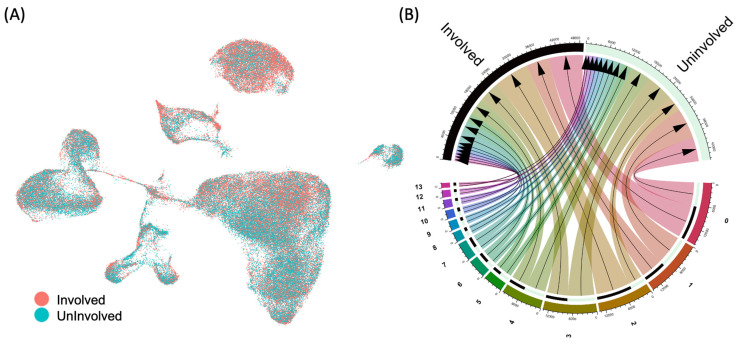
Cellular differences (**A**) and cluster differences (**B**) using the UMAP dimensionality reduction technique between the inflamed colonic tissue (Involved) and normal colonic tissue (Uninvolved) during anti-TNF therapy in CD.

**Figure 3 ijms-24-14099-f003:**
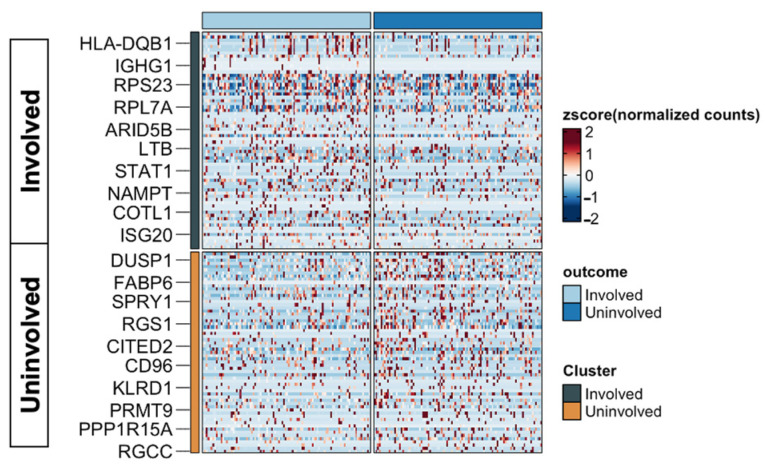
Heatmaps of commonly dysregulated genes associated with anti-TNF therapy resistance in Crohn’s disease. The light blue block and dark blue block in upper strata represent samples from inflamed colonic tissue (Involved) and normal colonic tissue (Uninvolved), respectively; the top 10 DEGs for each cluster and their corresponding normalized expression are shown in rows; the black coloured block and orange coloured block in the left side represent the results from the unsupervised clustering.

**Figure 4 ijms-24-14099-f004:**
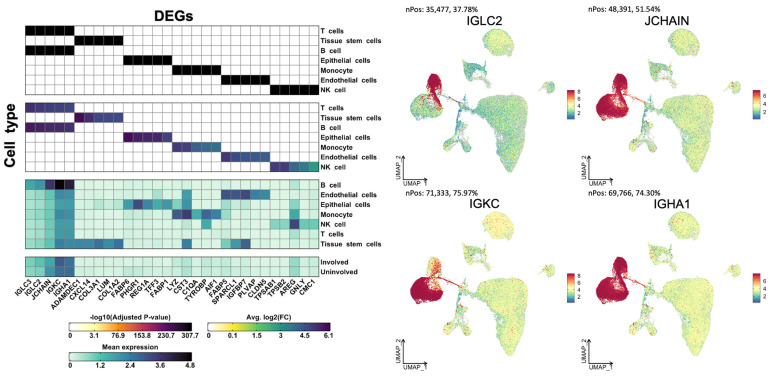
Differentially expressed genes by cell types for different levels of anti-TNF responsiveness.

**Figure 5 ijms-24-14099-f005:**
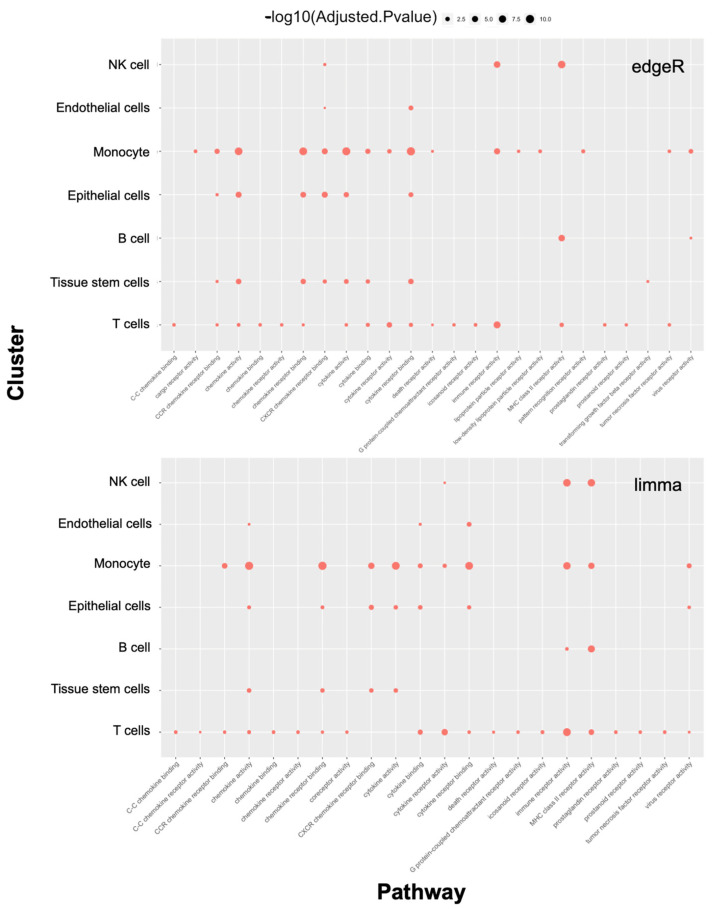
Representative GO terms of anti-TNF resistance-related differentially expressed genes by both edgeR and limma algorithms. Dot size represents the gene counts.

**Table 1 ijms-24-14099-t001:** Cellular coverage and drug score of vorinostat for anti-TNF-resistant CD.

Cell Type	Coverage within Cell Type	Drug Score	FDR
Limma			
Endothelial cells	1.59	56.23	<0.0001
Tissue stem cells	1.93	56.23	<0.0001
Epithelial cells	1.67	56.23	<0.0001
T-cells	51.04	56.23	<0.0001
EdgeR			
Endothelial cells	1.59	54.56	0.0002
T-cells	51.04	54.56	0.0009
Tissue stem cells	1.93	54.56	0.0015

Abbreviation: FDR, false discovery rate.

## Data Availability

The data that support the findings of this study are derived from publicly available sources (Gene Expression Omnibus database, http://www.ncbi.nlm.nih.gov/geo/).

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
