# Peer review of "Single-Cell Network-Based Drug Repositioning for Discovery of Therapies against Anti-Tumour Necrosis Factor-Resistant Crohn’s Disease"

_ijms, 2023, doi:10.3390/ijms241814099_

Round 1

Reviewer 1 Report

Abstract

  • In the background, it would be better to address why the authors re-analyzed the single-cell sequencing data in CD. 
  • In the method, the authors might want to provide the dataset information - GSE134809

Results

  • In the legends, all figures don’t have any explanation. The authors need to elaborate on each figure. 
  • The abbreviation should be mentioned in a full name for the first time (e.g. PCA).
  • In Figure 1A, the authors should assign the cell type in each cluster. I couldn’t find the legend in a graph. The authors would better mention each cluster’s annotation in A. Figure 1B might be deleted here.
  • In Figure 2, the authors must explain the meaning of ‘Involved’ vs ‘Uninvolved’. 
  • In Figure 2C, these two graphs should be separated. 
  • In Figure 3, the authors need to explain precisely. 
  • In Figure 4, IGLC2 seems like involved in endothelial cells. 
  • In Figure 4, I am not sure the color difference between ‘Involved’ and ‘Uninvolved’.
  • In supplementary figure S1B, every patient showed the same predicted result in the drugs. The authors should reanalyze the data and check that the result is correct. 
  • Overall, the figures have no connection. The explanation should be improved to make readers understand. The rigorousness should be considered. 
  •  

Extensive editing of English language required.

Author Response

Reviewer 1

In the background, it would be better to address why the authors re-analyzed the single-cell sequencing data in CD.

Responses]

Thank you very much for reviewing our paper.

As the reviewer’s comment, we have revised the abstract section. (See page 2 line 43):

“To date, very few single-cell researches exist for drug repurposing in Crohn’s disease.”

In the method, the authors might want to provide the dataset information - GSE134809

Responses]

Thank you very much.

As the reviewer’s comment, we have added the data access number.

In the legends, all figures don’t have any explanation. The authors need to elaborate on each figure. Responses]

Thank you very much for your kind responses about our paper.

We have changed as suggested.

The abbreviation should be mentioned in a full name for the first time (e.g. PCA).

Responses]

Thank you very much for reviewing our paper.

As the reviewer’s comment, we have revised the method section using full term of abbreviation. (See page 5 line 23):

“.. an average log fold change (log FC) larger than 0.25, and adjusted P values less than 0.05.”

Otherwise, we have already mentioned as the full name of abbreviation at the first time.

CD: page 4, line 2

TNF-α: page 4, line 7

PCA: page 5, line 15

UMAP: page 5, line 16

t-SNE: page 5, line 17

DEGs: page 5, line 21

GO: page 6, line 2

LINCS: page 6, line 8

However, as the reviewer knows, there’s no absolute consensus on what constitutes a “well-known” acronym, such as FDR. Therefore, we did not use full-term.

In Figure 1A, the authors should assign the cell type in each cluster. I couldn’t find the legend in a graph. The authors would better mention each cluster’s annotation in A. Figure 1B might be deleted here.

Responses]

Thank you very much for your kind responses about our paper.

We have changed the figure 1 as suggested, however did not deleted figure 1B, because assignment of cell type labels from clusters was showed to improve reading comprehension of readers.

Thank you for your detailed review again.

In Figure 2, the authors must explain the meaning of ‘Involved’ vs ‘Uninvolved’.

Responses]

Thank you very much for your good comment.

As the reviewer commented, we have changed description of the figure 2. (See page 20, line 6):

“Figure 2. Cellular differences (A) and cluster differences (B) using the UMAP dimensionality reduction technique between the inflamed colonic tissue (Involved) normal colonic tissue (Uninvolved) during anti-TNF therapy in CD.”

In Figure 2C, these two graphs should be separated.

Responses]

Thank you very much.

We have changed the figure 2 as suggested. (See Figure S1)

In Figure 3, the authors need to explain precisely.

Responses]

Thank you for this good point.

Based on the comments made by the reviewer, we have revised the manuscript. (See page 20, line 9):

“Figure 3. Heatmaps of commonly dysregulated genes associated with anti-TNF therapy resistance in Crohn’s disease. The light blue block and dark blue block in upper strata represent samples from inflamed colonic tissue (Involved) and normal colonic tissue (Uninvolved), respectively; the top 10 DEGs for each cluster and their corresponding normalized expression are shown in rows; the black colored block and orange colored block in the left side represent the results from the unsupervised clustering.’

In Figure 4, IGLC2 seems like involved in endothelial cells.

Responses]

Thank you for your detailed review.

Custer 5 was defined as B-cell in the specific functional annotations of cell clusters.

To better readability, we revised figure 1.

Thank you for your comment, again.

In Figure 4, I am not sure the color difference between ‘Involved’ and ‘Uninvolved’.

Responses]

Thank you very much for your kind responses about our paper.

We humbly accept your opinions.

Although the difference in color between the two groups was subtle and did not appear clear to the gross view in some genes, there were meaningful differences, and some genes had clear differences even.

In supplementary figure S1B, every patient showed the same predicted result in the drugs. The authors should reanalyze the data and check that the result is correct.

Responses]

Thank you for this good point.

As you commented, using limma method, the drug scores were similar across the patients, unlike using edgeR method.

Because the number itself is a small value, it appears similar on the graph, and the thing is that the candidate drug commonly recommended through different methods is vorinostat.

Overall, the figures have no connection. The explanation should be improved to make readers understand. The rigorousness should be considered.

Responses]

Thank you for valuable comments.

We humbly accept your opinions, and revised the manuscript.

We thank the reviewer for the careful reading of the manuscript and constructive remarks.

Extensive editing of English language required.

Responses]

Thank you for valuable comments.

We edited the manuscript through the premium English-editing service and attached the certification.

Reviewer 2 Report

The work is interesting, and certainly requires additional research, as this study indicates the involvement of B cells is particularly sensitive to HDACi modulation. In vitro, HDACi are capable of altering various aspects of B- a cellular response in a HDAC class-dependent manner that is maintained in both T-cell inactivation and T-cell-independent activation. The authors highlighted the regulation of cell type-specific genes at different levels of anti-TNF response by analyzing the single-cell transcriptome ysis, which allows detailed analysis of gene expression variation between cells. Repurposing a drug based on compatible genes shows that promising drugs for repurposing are more likely to be identified, allowing development of personalized CD therapy. Nevertheless, that study has some limitations. First, the sample size is relatively small, so a dropout experiment method within the algorithm was  applied to calculate the drug score . Secondly the autors  findings alone in silico,  findings alone, without experimental validation, do not fully explain the biological mechanisms underlying the responsiveness to anti-TNF-α therapy in patients with CD.

Author Response

Dear Editor,

We appreciate the opportunity to revise our work for consideration for publication in International Journal of Molecular science. We hope our revision meet your approval. We next detail our responses to each reviewer’s concerns and comments.

Reviewer 2

Suggestions for Authors

The work is interesting, and certainly requires additional research, as this study indicates the involvement of B cells is particularly sensitive to HDACi modulation. In vitro, HDACi are capable of altering various aspects of B- a cellular response in a HDAC class-dependent manner that is maintained in both T-cell inactivation and T-cell-independent activation. The authors highlighted the regulation of cell type-specific genes at different levels of anti-TNF response by analyzing the single-cell transcriptome ysis, which allows detailed analysis of gene expression variation between cells. Repurposing a drug based on compatible genes shows that promising drugs for repurposing are more likely to be identified, allowing development of personalized CD therapy. Nevertheless, that study has some limitations. First, the sample size is relatively small, so a dropout experiment method within the algorithm was  applied to calculate the drug score . Secondly the autors  findings alone in silico,  findings alone, without experimental validation, do not fully explain the biological mechanisms underlying the responsiveness to anti-TNF-α therapy in patients with CD.

Responses]

Thank you very much for your kind responses about our paper.

We humbly accept your opinions, and now plan the further experimental work.

We thank the reviewer for the careful reading of the manuscript and constructive remarks.

Round 2

Reviewer 1 Report

All comments were properly revised.

Moderate editing of English language required